# Research on Elevation Survey Method of Sea-Crossing Bridge under Adverse Conditions

**Jun Xiao** [1,2], **Jianping Xian** [1,2], **Song Li** [1] and **Shuai Zou** [2,3,*]

1   CCCC Second Highway Engineering Co., Ltd., Xi'an 710065, China
2   Shanxi Union Research Center of University and Enterprise for Bridge Intelligent Construction, Xi'an 710199, China
3   School of Civil Engineering, Chongqing Jiaotong University, Chongqing 400074, China
*   Correspondence: zs2448602237@163.com; Tel.: +86-186-8169-2538

**Abstract:** Aiming to survey scenarios of offshore projects with difficult horizontal elevation transmission and long-distance, all-weather elevation monitoring operations, a long-distance, total station, trigonometric leveling based on dynamic compensation is proposed. The feasibility of this method was verified by an outdoor survey experiment, and the range of transverse coverage and accuracy reached by this method was quantitatively analyzed. The results indicate that this method shows a good correction effect on the survey results of test points under different environmental conditions, which proves that this method is feasible. The correction effect of this method is affected by the distance between the test point and the datum point; within the range of 60 m horizontally from the datum point, an assurance rate of about 90% can be achieved for the error range of 20 mm. Combining with the built-in ATR (Automatic Target Recognition) technology of the total station, this method can make the elevation survey result reach the accuracy of millimeter level under the range of about 1000 m, by obtaining multiple groups of data and then calculating the mean value. This paper provides a new method for the elevation transfer of sea-crossing bridges under long-distance conditions and harsh environmental conditions.

**Keywords:** sea-crossing bridges; triangle elevation; leveling method; dynamic compensation; total station; harsh environment; long distance





## 1. Introduction

In recent years, there has been an upsurge in sea-crossing traffic engineering construction around the world, and the length of sea-crossing bridges has been refreshed again and again. The earliest sea-crossing bridge in China is the Xiamen Bridge, which was built in 1987. The total length of the bridge is 6.695 km. It was finally completed in 1991 and opened to traffic in May. Since then, the curtain of China's sea-crossing bridge construction has been lifted [1]. The following Table 1 lists the top 10 typical sea-crossing bridges that have been built or are under construction in China.

To meet the increasing traffic demand, the construction of the sea-crossing bridge is gradually developing towards the strong-wave, deep-water area. Its length ranges from several kilometers to tens of kilometers. Large-scale sea-crossing projects under harsh environmental conditions will inevitably cause many severe technical problems in construction and maintenance [2,3]. In the maintenance of large-scale sea-crossing bridges, to meet the high-standard testing requirements, it has become an important research direction in this field to innovate the testing methods by combining the traditional non-destructive testing (NDT) methods with advanced techniques [4,5]. Khedmatgozar Dolati et al. [4] compiled in one place all the NDT techniques, including the application of drones, sensors, or robots for rapid and efficient assessment of damage on small and large scales, which kept researchers up-to-date with existing methods and paved the way for further innovations in this regard. In the construction of sea-crossing bridges, long-distance

elevation surveys in a complex marine environment is one of the important technical problems. In the project, some offshore survey platforms are often established through offshore test pile projects, and some bridge piers are under priority construction so that the survey distance can be shortened to less than 2 km, which provides practical conditions for using high-precision total stations to transfer the elevation data to offshore buildings or structures and to realize the direct connection of the sea-crossing elevation [6,7]. Affected by waves and fluctuating tides, the offshore survey platform and piers are always subject to slight shaking, and the vertical angle of the total station placed on the platform or pier can be changed for tens of seconds or even minutes [8,9]. To ensure the survey effect, the stiffness and size of the offshore survey platforms are often designed to be very large, and the distance between the survey platforms will be increased based on the consideration of construction cost.

**Table 1.** The top 10 typical sea-crossing projects that have been built or are under construction in China.

| Order | Bridge Name | Span/km | Built/Building |
| --- | --- | --- | --- |
| 1 | Hong Kong–Zhuhai–Macao Bridge | 55.0 | Built |
| 2 | Jiaozhou Bay Bridge | 36.5 | Built |
| 3 | Hangzhou Bay Bridge | 36.0 | Built |
| 4 | East Sea Bridge | 32.5 | Built |
| 5 | Quanzhou Bay Bridge | 26.7 | Built |
| 6 | Shenzhen–Zhongshan Bridge | 24.0 | Building |
| 7 | Jintang Bridge | 18.5 | Built |
| 8 | Huangmao Hai Link—Cross-sea section | 14.4 | Building |
| 9 | Jiashao Bridge | 10.1 | Built |
| 10 | Nan'ao Bridge | 9.3 | Built |

For high-precision total station trigonometric leveling, the main factors affecting the accuracy of the survey are the size of the zenith distance, the observation error, and the measurement error of the vertical atmospheric refraction coefficient, if the stability of the station is guaranteed [10,11]. The influence degree of each factor gradually increases with the increase in the distance between the survey station and the target prism. Therefore, it is of great significance to improve the accuracy of the survey by either improving the survey method or compensating and correcting the survey results to reduce the deviation caused by various influencing factors the survey results.

Qiu Yang et al. [12] used auxiliary equipment to measure the height of the total station and the target plate, appropriately increased the number of angular survey rounds to improve the accuracy of vertical angle, and carefully planned the survey steps to reduce the vertical atmospheric refraction error and other means to effectively eliminate and reduce the impact of the main error source. Yaming Xu et al. [13] proposed a linear structure survey method based on the traditional method, combined with relatively strict synchronous opposite observation, which reduced the complexity and observation amount of the original method. References [14–16] improved and optimized the survey method proposed in reference [13] and carried out long-distance, sea-crossing elevation transmission in the Chuandao area of Jiangmen Taishan City. Zhongping Wang et al. [17] improved the point layout structure based on the trigonometric elevation observation method proposed in reference [13] and eliminated and screened the data of each survey section in combination with the specification and reasonable restriction of the elevation difference between each group. The requirements of the national second-class leveling accuracy were met through project testing.

Heng Zhang et al. [18] have greatly weakened the influence of atmospheric refraction, vertical deviation, and Earth curvature by using two high-precision total stations of the same model to observe the opposite direction of the high–low biprism group at the same time. Based on the least square theory, Mingbo Liu et al. [19] gave a new method to calculate the refraction coefficient of the survey area. It was demonstrated in production practice that using this method can produce triangular elevations with accuracy close to

the second-class level. Jianzhou Li et al. [20] deduced the error formula of the survey result through the error propagation law according to the rigorous calculation formula of precision trigonometric leveling and proved that controlling the vertical angle and survey distance of the instrument can improve the survey accuracy and achieve second-class leveling accuracy. Jianzhou Li et al. [21] developed TriLevel, a precision triangulation elevation survey system that realizes the automatic measurement and quality control of precision triangulation elevation. Refs. [22–24] weakened the deviation caused by atmospheric refraction on elevation survey results by avoiding measuring the total station height and realizing a high-precision elevation survey. Peibing Yang et al. [25] designed a special observation mark to solve the problem of long-distance surveys, aiming at the need for high-precision elevation transmission in wide sea areas, and also responded to the impact of the shaking of the survey platform on the survey accuracy by carrying out rigorous and rapid observation operations under favorable survey conditions.

At present, there are many pieces of research on the improvement of the survey method of total station trigonometric leveling and the methods of compensation and correction of the survey results, which can make the accuracy of the survey results reach a high level, but to ensure the accuracy requirements of the survey results, the improved survey method or the method of correction of the survey results are generally cumbersome. However, there is relatively little research on developing targeted, simple, and efficient survey methods or correction methods under some specific conditions and harsh environments.

Because of this, this paper proposes a long-distance total station elevation survey method based on dynamic compensation, which is aimed at survey scenarios of offshore projects with difficult horizontal elevation transmission under long-distance, all-weather elevation monitoring operations and harsh environmental conditions. The feasibility of the proposed elevation survey method was verified by an outdoor survey experiment. At the same time, through the processing and analysis of a large number of surveyed data at different test points, a quantitative study was carried out within the scope that this method can cover in the horizontal direction and the accuracy that the survey results can reach. This method can be used as an efficient and high-precision method for the elevation survey of sea-crossing bridges under adverse conditions.

## 2. Principles and Methods

### 2.1. Engineering Background

The elevation survey method proposed in this paper mainly serves the elevation survey operation in the construction of the Peljesac Bridge. The Peljesac Bridge is located at the southern end of Croatia, bordering Bosnia and Herzegovina. It is located in the Malostonski Bay protection area on the west side of the Adriatic Sea, with the Peljesac Peninsula and mainland on either side of the bridge. The main Peljesac Bridge is a low-tower, cable-stayed bridge with a total length of 2404 m, which has a central single cable plane and a steel box girder, and its span group is (84 + 108 + 108 + 189.5 + 5 × 285 + 189.5 + 108 + 108 + 84) m. The bridge has 14 piers and abutments, of which 4 piers and abutments are arranged on land on both banks, and the remaining 10 pier towers are in the water. The layout of the Peljesac Bridge is shown in Figure 1.

During the construction of the Peljesac Bridge, no sea survey platform was set up. By setting survey stations on both banks, the elevations of the bridge piers and tower columns in the sea were surveyed. The survey operations are shown in Figure 2.

### 2.2. Basic Principle of Total Station Trigonometric Leveling

Trigonometric leveling is a method to determine the height difference between two points by surveying the horizontal distance and vertical angle of the two control points. The basic formula for the calculation of one-way survey height difference is shown in Formula (1), which is generally derived from Figure 3:

$$h_{AB} = S cos\alpha + \frac{1-K}{2R}S_0^2 + i_A - v_B \tag{1}$$

where $S$ is the oblique distance between two points $A$ and $B$, m; $\alpha$ is the vertical angle between the observation point of the total station and the target prism, (°); $R$ is the average curvature radius of the earth, $R \approx 6371$ km; $K$ is the local atmospheric refraction coefficient (related to temperature, pressure, and atmospheric density), which can be taken as 0.10~0.15 for general mountainous areas and 0.5 for flat areas near the ground; $S_0$ is the horizontal distance between two points, m, $S_0 = S\cos\alpha$; $i_A$ is the height of the total station, m; $v_B$ is the height of the prism, m.

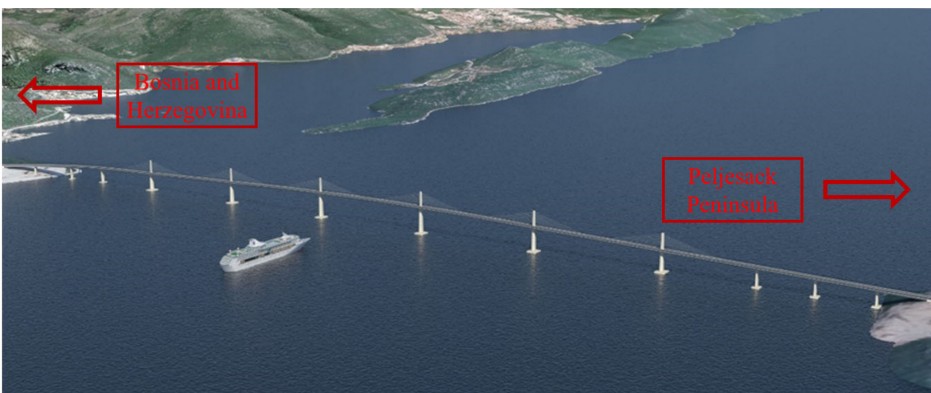

**Figure 1.** The layout of the Peljesac Bridge.

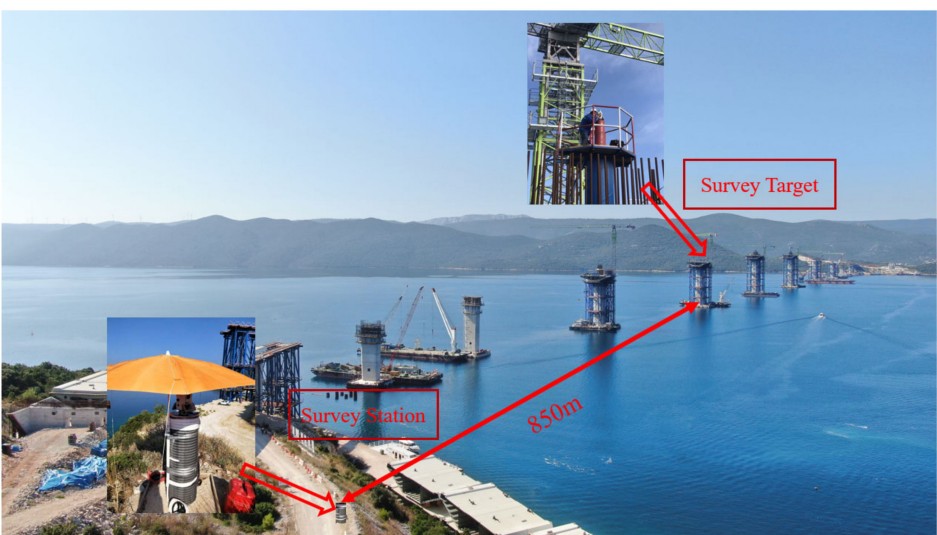

**Figure 2.** The process of survey operations.

The calculation formula of the mean square error in the elevation difference of a one-way survey can be derived from the total differential of Formula (1) [26] and then from the error propagation rate, as shown in Formula (2):

$$M_{h_{AB}}^2 = S^2 \cos^2\alpha \frac{m_\alpha^2}{\rho^2} + \sin^2\alpha\, m_s^2 + \frac{1}{4R^2} S_0^4 m_{K_i}^2 + m_{i_A}^2 + m_{v_B}^2 \qquad (2)$$

where $m_\alpha$ is the mean square error of vertical angle observation; $m_S$ is the mean square error of the oblique distance survey; $m_K$ is the mean square error of the atmospheric refraction coefficient measurement, generally $\pm 0.03 \sim \pm 0.05$ [27]; $m_{i_A}$ is the mean square error of the total station height measurement; $m_{v_B}$ is the mean square error of the target height measurement; $\rho$ is the constant used for angle unit conversion, $\rho = 206265$.

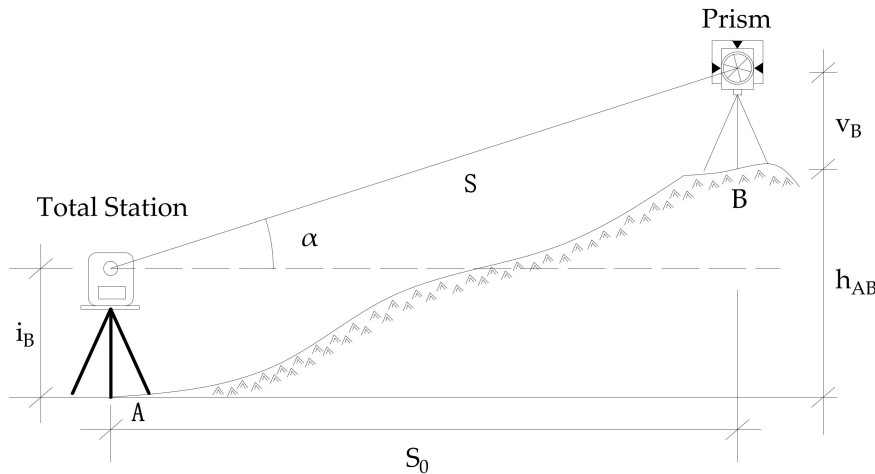

**Figure 3.** Principle of triangle elevation.

### 2.3. Principle of Dynamic Compensation

According to the calculation formula of the mean square error of the survey height difference shown in Formula (2), the high-precision total station needs to be compensated and corrected due to the influence of atmospheric refraction, topographic conditions, the earth's curvature, etc., during long-distance trigonometric leveling. As shown in Figure 4, the line of sight $e1$ of the datum point in the figure and the line of sight $e2$ of the test point to be surveyed should be the same at the same time because the distance between the datum point and the test point is short.

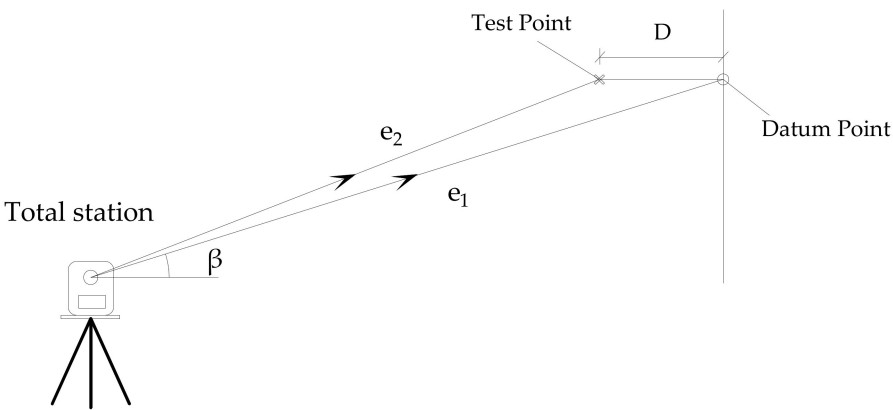

**Figure 4.** The principle of dynamic compensation.

If the accurate elevation data of the datum point is known, the elevation can be transmitted to the datum point by densifying the datum point on land or structure under the meteorological conditions required in the survey specification, such as night or morning with good weather conditions, and finally determined by taking the mean value through multiple surveys; when the survey operation conditions allow, the accurate elevation value of the datum point can also be obtained through leveling. When the elevation survey operation is carried out, the elevation of the datum point is surveyed first, and the difference between its accurate elevation value and the current elevation survey value is taken as the dynamic compensation correction value under the current condition. Then, the elevation of the test point is surveyed, and the obtained dynamic compensation correction value is used to correct the elevation survey result of the test point, to reduce or eliminate the influence of external factors. The correction formula is as follows:

Dynamic compensation value:

$$\mu = H_0 - H_0' \tag{3}$$

Correction value of current survey result:

$$H_i' = H_i + \mu \tag{4}$$

where $H_0$ is the precise elevation value of the datum point; $H_0'$ is the elevation survey result of the datum point during the current survey; $\mu$ is the dynamic compensation value of the current elevation survey result; $H_i$ is the elevation survey result of the point to be surveyed in the current survey; $H_i'$ is the elevation correction value of the point to be surveyed in the current survey.

### 2.4. Introduction of Experiment

To verify and carry out quantitative research on the trigonometric leveling method based on dynamic compensation proposed in this paper, an outdoor field survey experiment was carried out. The specific contents of the experiment are as follows.

### 2.4.1. Scheme Design

Considering the ultimate purpose of this experiment, the selection of the experimental site should meet the following requirements:

(1) The distance between the survey station and the datum point should be far enough; there should be no obstructions within the line of sight between the survey station, the datum point, and the test point; and the intervisibility condition should be good.
(2) The survey site should allow the convenient use of the level gauge to transmit the elevation of the survey station to the datum point and the test point station by station;
(3) A certain range around the datum point should be open, which is convenient for the layout of the test points under different conditions.

To meet the requirements of the above experimental site, an open construction site was selected for this experiment, and the layout of the site is shown in Figure 5. To quantify the range that the method can cover horizontally over long distances, the distance between the datum point (JZ01) and the survey station (CZ01) was selected as 1000 m. According to the different distances between the datum point and the test point, a total of seven test points were proposed, as shown in Table 2. Due to the long-distance survey, the efficiency of the target prism aiming manually is low, and large observation errors may be caused by human factors. Therefore, the built-in ATR function of the total station was used for target collimation in this experiment. At the same time, to verify the applicability of this method under different environmental conditions, this experiment chose to continuously observe for a day, and the observation frequency was 1 group every 10 min.

**Table 2.** Test point details.

| Test Point | Distance from Datum Point L/m | Collimation Mode | Measurement Frequency |
|---|---|---|---|
| DC-30 | 30 | ATR | |
| DC-50 | 50 | ATR | |
| DC-70 | 70 | ATR | |
| DC-60 | 60 | ATR | 1 Group/10 Min |
| DC-80 | 80 | ATR | |
| DC-90 | 90 | ATR | |
| DC-100 | 100 | ATR | |

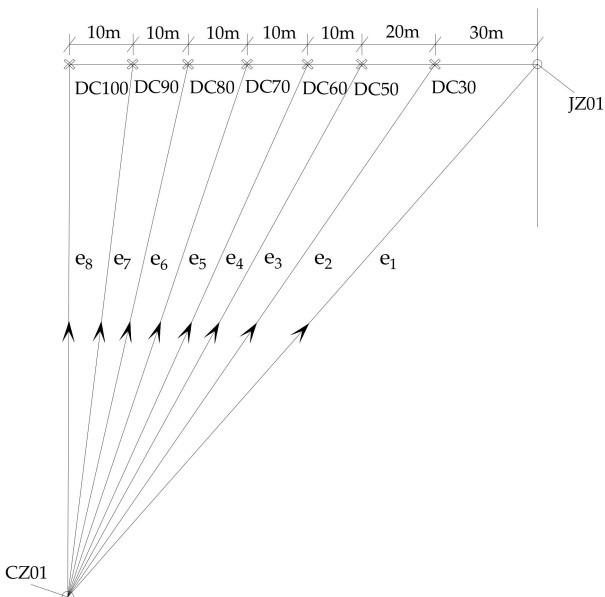

**Figure 5.** The layout of the site.

### 2.4.2. Experiment Content

In order to carry out the experiment orderly and ensure the reliability of the survey results, the experiment should be carried out according to the following operating steps:

(1) In the night or morning with good weather conditions, the elevation of the survey station should be transmitted to the datum point and the test points station by station by using the precision level, and the surveyed elevation values should be drawn up as the accurate elevation values of each point.

(2) During the actual survey, set up a total station at the station, input the current environmental parameters (such as temperature, air pressure, humidity, etc.) into the total station for preliminary correction, and make corresponding records at the same time;

(3) Survey the elevation value of the datum point first, and take the difference between this survey value and the accurate elevation value of the datum point surveyed in step (1) as the correction value of the current survey (errors caused by atmospheric refraction, earth curvature, etc.).

(4) Survey the elevation values of the test points under each working condition in turn, and correct the elevation survey results of each test point by using the correction value in step (2) according to Formula (4) in Section 2.3.

(5) Repeat steps (2), (3), and (4) within a fixed time interval to continuously survey the elevation of the test points.

The test procedure is shown in Figure 6.

The following points should be paid attention to during the experiment to ensure the accuracy of the experimental results.

a. In step (1), the accurate elevation values of the datum point and the test points are very important for the correction of the survey results and the evaluation of the experimental results. Therefore, the accuracy of the accurate elevation value survey of each point should be guaranteed.

b. As the environmental conditions in a day are changing at all times, the corresponding parameters (temperature, air pressure, humidity, etc.) should be input into the total station for preliminary correction before each data observation.

c. To ensure that the survey process is completed under basically the same environmental conditions, the survey results of each group should be completed within two minutes.

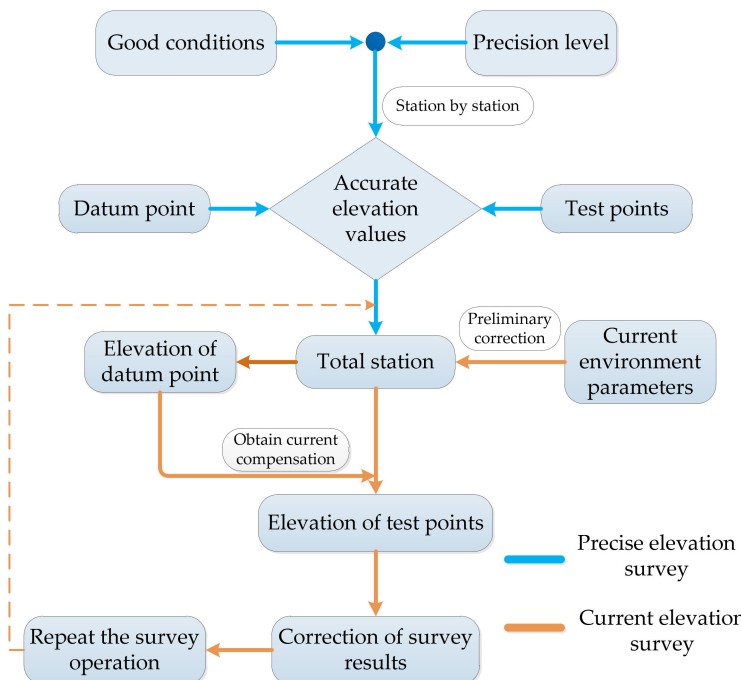

**Figure 6.** The procedure of the outdoor survey test.

### 2.4.3. Experimental Equipment

The orderly conduct of the experiment is closely related to the preliminary preparations. Therefore, the test equipment shall be prepared in advance and shall be within the qualification period. The equipment required for this experiment includes a high-precision level, a high-precision total station (Leica TS60), a thermometer, a barometer, a humidity meter, some measuring nails, a manual painting, a test record book, etc. A Leica TS60 total station with an angle survey accuracy of 0.5″ and a distance survey accuracy of 0.5″ was used for this experimental survey, with a survey range of 3500 m in ATR mode, to ensure the testing effect.

Before the experiment, the specific position of the datum point and each test point should be determined through the distance survey function of the observation instrument, and the datum point and the test points should be marked by measuring nails and manual painting.

### 3. Results

Because of the constant changes in external factors (temperature, atmospheric refraction, and so on), the results of each survey varied to varying degrees. According to the basic principle of the elevation survey method proposed in this paper, the difference between the elevation survey result of the datum point and the accurate elevation value of the datum point was taken as the current dynamic compensation value $\mu$, which was used to correct the elevation survey results of each test point to be surveyed at that time.

Through the analysis of multiple groups of data at different test points, the assurance rate of different survey accuracies and the overall correction effect is used to evaluate the applicability of the method under different distances between the datum point and the test point.

For simplicity of expression, the following applicable scope refers to the scope covered by the distance between the datum point and the test point, which can be used to make a relatively ideal correction of the elevation survey results. This scope is the circular area with the datum point as the center and the distance between the datum and the test point as the radius.

*3.1. Experimental Result*

According to the basic principle of the elevation survey method proposed in this paper, the accuracy of the survey results was evaluated by the following Formula (5):

$$\Delta H = \left(H_i' - H_{i0}\right) \times 1000 \tag{5}$$

where $\Delta H$ is the deviation between the corrected survey result and the precise elevation value of the test point, mm; $H_{i0}$ is the precise elevation survey value of the test point, m; and $H_i'$ is the elevation correction value of the test point in the current survey, m.

Figure 7 shows the changes in accuracy curves before and after the correction of elevation survey results for each test point at different times. The figure shows:

(1)　From the uncorrected data results of each test point in each figure, it can be seen that the elevation survey results directly using the total station are greatly affected by the environmental conditions at different times, showing an obvious survey window period. In the early morning with good weather conditions, the light, humidity, temperature, and intervisibility conditions are all within the scope specified in the survey, and the deviation between the survey results and the accurate value is relatively small. Under the ranging condition of about 1000 m, the maximum deviation between the survey result and the accurate value is about 24 mm. With the change in environmental conditions, the deviation between the surveyed results and the accurate value gradually increases.

(2)　When the elevation is continuously surveyed in a day, the effect of the continuous change in external environmental conditions on the total station can cause the survey results to shift in a certain direction. Under the experimental conditions described in this paper, the maximum deviation between the elevation survey results and the accurate value can even reach about 25 cm for a test point, and at twilight, when the environmental conditions are relatively mild, the effect of the previous environmental conditions on the total station will continue to affect the elevation survey results. After the survey results are corrected, they can return to the accurate value.

(3)　From the variation in the deviation between the survey results of each test point in the figure before and after correction and the accurate value, it can be seen that each test point at different distances from the datum point, after the elevation survey method proposed in this paper is used to correct the elevation survey results at any time, the corrected elevation results can fluctuate within a certain range near the accurate value. At the test point, which is 30 m away from the datum point, the deviation between the corrected survey result and the accurate value can be kept within 20 mm.

(4)　Under the same environmental conditions, with the increase in the distance between the test point and the datum point, the fluctuation range of the corrected survey results generally shows a gradually increasing trend. At the test point at a distance of 30 m from the datum point, the maximum deviation between the corrected elevation survey result and the accurate value is 18.0 mm; at the test point at a distance of 100 m from the datum point, the maximum deviation between the corrected elevation survey result and the accurate value is 34.5 mm.

*3.2. Data Process*

3.2.1. The Accuracy Assurance Rate

The quantitative statistical analysis of the survey accuracy of multiple groups of survey data at the same test point can be completed by the assurance rate under the corresponding accuracy conditions, and the accuracy assurance rate of survey results $\eta$ is calculated with the following formula:

$$\eta = \frac{N}{n} \times 100\% \tag{6}$$

$$N = number\left(|\Delta H| \leq \alpha\right) \tag{7}$$

or

$$N = number(|\Delta H| > \alpha) \tag{8}$$

where *n* is the number of s data groups; *N* is the number of groups for which the absolute value of the the survey result's correction value accuracy meets a certain condition; and *α* is the accuracy condition of the survey results, mm.

Figures 8 and 9 show the distribution of different accuracy assurance rates of the corrected survey results and the variation trend of the same accuracy ratio with the increase in the distance between the test point and the datum point, respectively.

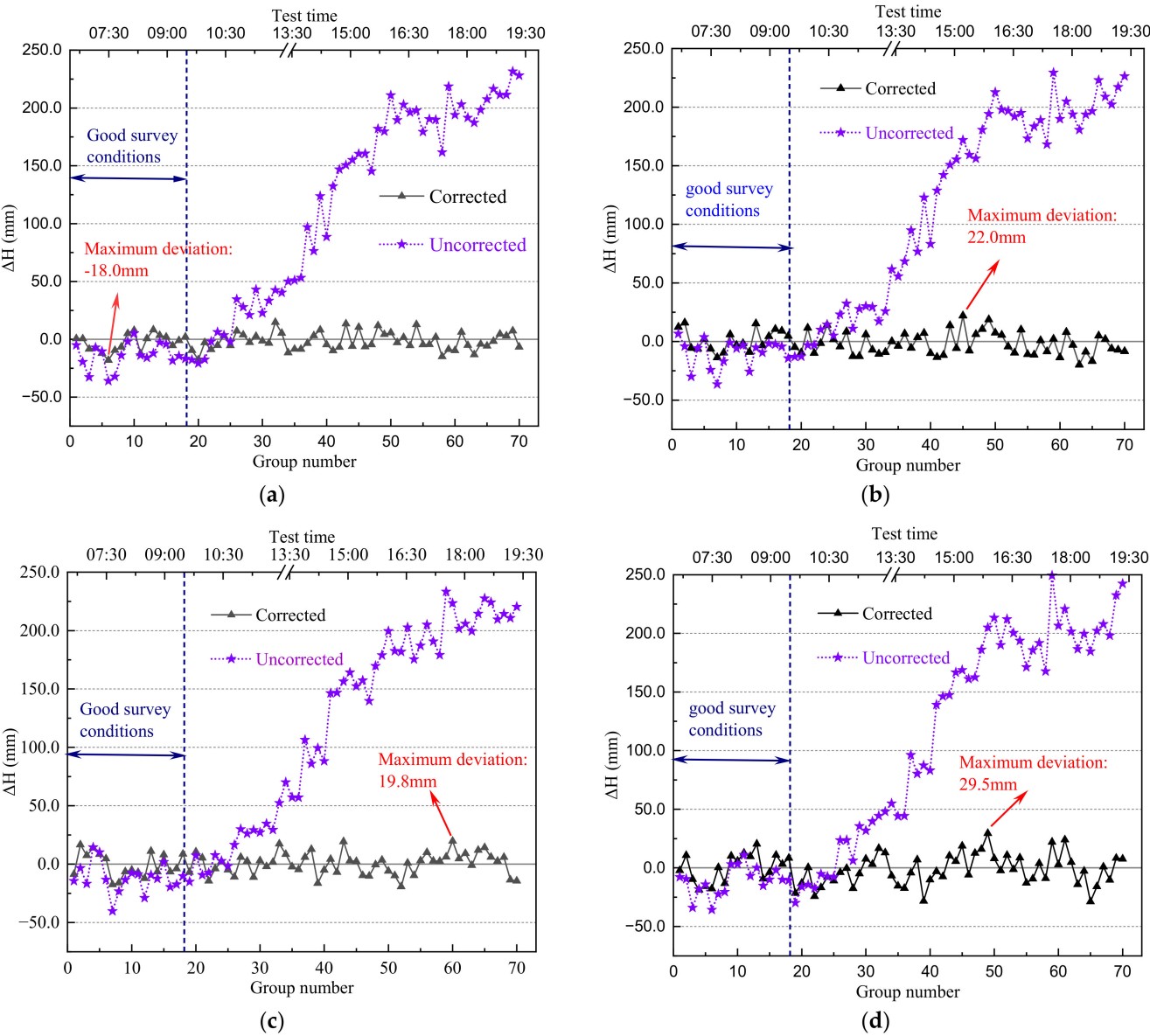

**Figure 7.** *Cont.*

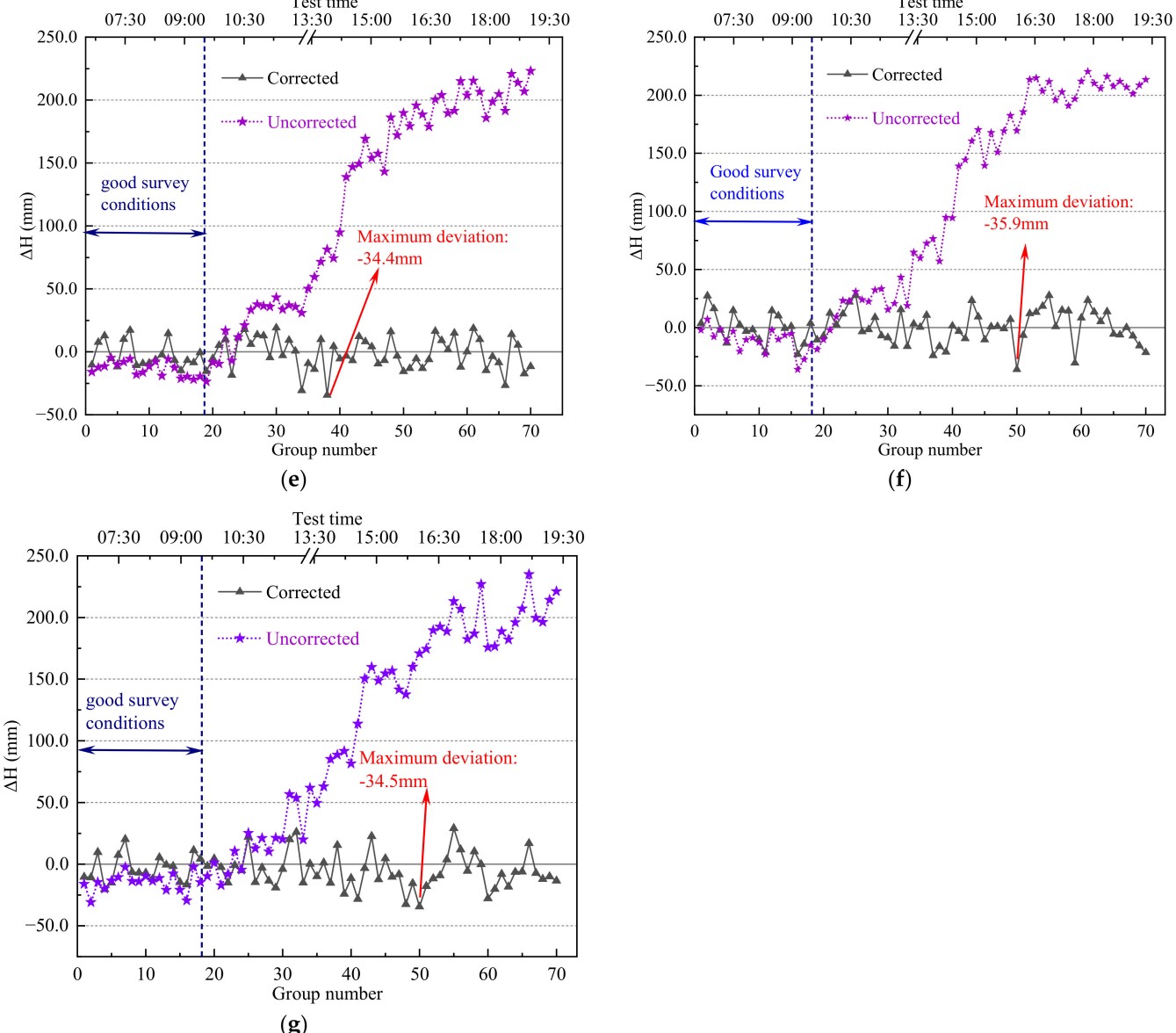

**Figure 7.** The curves of accuracy change: (**a**) DC-30; (**b**) DC-50; (**c**) DC-60; (**d**) DC-70; (**e**) DC-80; (**f**) DC-90; (**g**) DC-100.

Figures 8 and 9 show that:

a.  With the increase in the distance between each test point and the datum point, the accuracy assurance rate generally shows a downward trend under the same accuracy requirement.

b.  By correcting the survey results, within a 30 m horizontal distance from the datum point, the error range of 15 mm can reach a 97.2% assurance rate, and the error range of 20 mm can reach 100% assurance rate; within the transverse distance of 60 m from the datum point, the error range of 20 mm can reach about 90% assurance rate.

c.  With the increase in the distance between each test point and the datum point, the ratio of different accuracy conditions shows different trends. The ratio of accuracy conditions of 0~5 mm and 5~10 mm shows a decreasing trend. The ratio of 10~15 mm accuracy conditions remains unchanged, while the ratio of greater than 15 mm accuracy conditions shows an upward trend.

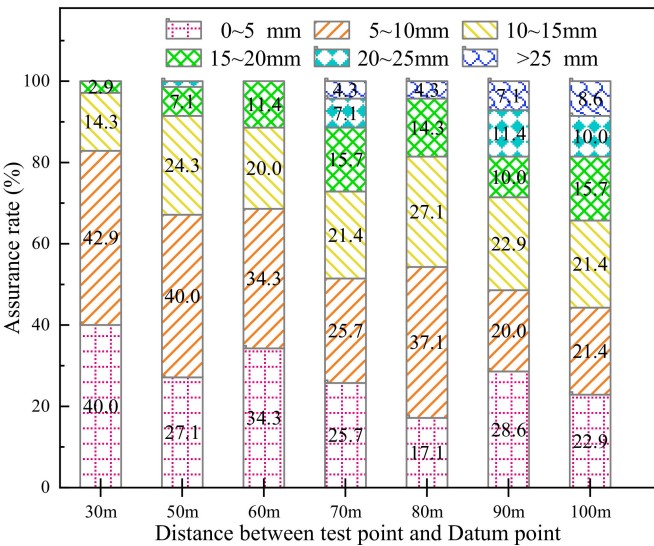

**Figure 8.** Accuracy assurance rate distribution.

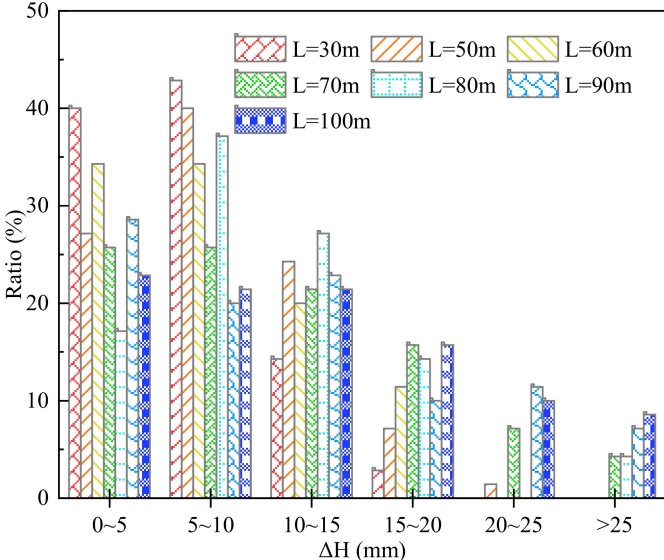

**Figure 9.** Accuracy ratio trend.

### 3.2.2. Data Dispersion

The mean value of the data sample reflects the centralized trend of the data. Table 3 shows the corrected mean value of multiple groups of survey results at all test points.

**Table 3.** The corrected mean value.

| Test Point | Distance from Datum Point L/m | Corrected Deviation Range/mm | Corrected Deviation Mean/mm | Remarks |
|---|---|---|---|---|
| DC-30 | 30 | −18.0~14.8 | −1.9 | |
| DC-50 | 50 | −19.8~22.0 | −1.0 | |
| DC-70 | 70 | −19.3~19.8 | 0.2 | The lowest temperature: is 21 °C; |
| DC-60 | 60 | −28.7~29.5 | −1.1 | The highest temperature: is 37 °C; |
| DC-80 | 80 | −34.4~19.1 | −1.5 | The data statistics of each test |
| DC-90 | 90 | −35.9~27.7 | 0.7 | point: 70 groups. |
| DC-100 | 100 | −34.5~29.1 | −5.0 | |

It can be seen from the data in Table 3 that after the elevation survey method provided in this paper is used to correct the survey results, the survey results can better approximate the accurate value by taking the mean value of multiple groups of corrected elevation data and finally reach millimeter-level accuracy. Moreover, the mean value of the corrected data is less affected by the distance between the test point and the datum point and the change in the survey environment.

## 4. Discussion

Because of its great convenience, the use of a high-precision total station for elevation surveys has been widely concerned and used by the engineering community. In a long-distance elevation survey, its accuracy is significantly affected by atmospheric refraction, the Earth's curvature, temperature, and other factors, and the survey results may have a large deviation [10,11]. It is therefore very necessary to compensate for and correct the survey results.

A method of conducting an elevation survey for long distances based on dynamic compensation is proposed in this paper, which makes use of the principle that the compensation amount of two points that are close to each other shall be the same in a short time to correct the elevation survey results. Through the above experimental analysis, this method had a good correction effect on the elevation survey results within the experimental range, and its correction effect was affected by the distance between the test point and the datum point, thus showing the change in the corresponding accuracy assurance rate under different distances. In the actual project, the elevation survey method proposed in this paper can be reasonably selected according to the accuracy requirements of the project. In the above experiments, the uncorrected total station elevation survey results showed an obvious window period for the impact of environmental changes. In the morning, when the environmental conditions were suitable, the survey results were relatively stable, and the survey accuracy was relatively high. Therefore, it is recommended to adjust the long-distance elevation survey to the morning or night with good weather conditions, to reach a more ideal survey effect. Since the elevation survey method can maintain good adaptability under any environmental conditions, if combined with the ATR function of the total station and the corresponding data-processing program for the total station is developed, the method of taking the mean value after multiple surveys can also be used as a high-precision elevation survey method suitable for harsh environmental conditions.

To sum up, the new method proposed in this paper for elevation surveys under adverse conditions can provide sufficient technical support for elevation surveys in the construction of large-scale sea-crossing bridges due to its efficiency, high precision, and strong environmental adaptability. Limited by space, this paper only presents experimental research on the application of this method under the condition of a 1000 m range and does not involve research on the accuracy that this method can achieve under different range conditions. The relevant research will be carried out in the follow-up to clarify the change rules for the applicability, coverage area, and accuracy of measurement results with the distance of range.

## 5. Conclusions

By experimenting, this paper verified the feasibility of a total station elevation survey method based on dynamic compensation. At the same time, it made a quantitative study on the horizontal coverage of this method and further analyzed the practical situation of this method by counting the accuracy of the survey results under the condition of different distances from the datum point. The main conclusions are as follows:

(1)     The analysis shows that the total station elevation survey method based on dynamic compensation proposed in this paper shows a good correction effect on the survey results within 100 m between the test point and the datum point, making the survey results fluctuate within a certain range near the accurate value, which proves that the survey method proposed in this paper is feasible.

(2) The accuracy of the corrected survey results is affected by the distance between the test point and the datum point. Through the correction of the survey results, the assurance rate of 97.2% can be reached in the error range of 15 mm and 100% in the error range of 20 mm within the range of 30 m horizontally from the datum point.

(3) By using this method to correct the survey results and combining it with the built-in ATR technology of the total station to obtain multiple groups of data, and then calculating the mean value, the elevation survey results can reach millimeter-level accuracy within the application range of 100 m from the datum point.

(4) The correction effect of the survey method provided in this paper on the elevation survey results is less affected by the environment. The method of taking the mean value after obtaining multiple groups of data can also be considered a high-precision elevation survey method suitable for harsh environmental conditions.

**Author Contributions:** Conceptualization, J.X. (Jun Xiao) and J.X. (Jianping Xian); methodology, J.X. (Jun Xiao), S.L. and S.Z.; validation, J.X. (Jun Xiao), J.X. (Jianping Xian) and S.L.; formal analysis, S.Z.; investigation, J.X. (Jun Xiao), S.L. and S.Z.; resources, J.X. (Jun Xiao); data curation, J.X. (Jun Xiao), J.X. (Jianping Xian) and S.Z.; writing—original draft preparation, J.X. (Jun Xiao), J.X. (Jianping Xian), S.L. and S.Z.; writing—review and editing, S.Z.; visualization, J.X. (Jun Xiao), J.X. (Jianping Xian) and S.L.; supervision, J.X. (Jun Xiao); project administration, S.Z.; funding acquisition, J.X. (Jun Xiao). All authors have read and agreed to the published version of the manuscript.

**Funding:** This research was funded by the Science and Technology Project of CCCC (grant number: No. 2020-ZJKJ-QNCX04) and the Science and Technology Special Major Project of CCCC (grant number: No. 2019-ZJKJ-07).

**Institutional Review Board Statement:** Not applicable.

**Informed Consent Statement:** Not applicable.

**Data Availability Statement:** Data are contained within the article.

**Conflicts of Interest:** The authors declare no conflict of interest.

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
