# Peer review of "Research on Elevation Survey Method of Sea-Crossing Bridge under Adverse Conditions"

_sustainability, doi:10.3390/su141811641_

Round 1

Reviewer 1 Report

Reviewer #1: This manuscript proposes a long-distance elevation survey method based on dynamic compensation, which give a useful method for the survey scenarios of offshore projects with difficult horizontal elevation transmission under long-distance, all-weather elevation monitoring operations and harsh environmental conditions. In general, it is an interesting investigation and will improve the efficiency and precision of the elevation survey under the condition of limited horizontal transmission of elevation. Therefore, the reviewer recommends that this manuscript will be accepted after the minor modifications.

The minor revisions are listed as follows:

(1) Line 116, the English expression needs to be improved.

(2) The detailed description of the specific time frame, completed of each group of tests, should be added in the manuscript of section 2.4.2.

(3) In Fig.6 (a); (b), (c), (d), (e), (f), (g), "good survey conditions" denoted by the blue texts, should give the detailed description.

(4) The notes of relevant formulas should be perfected in section 3.2.1. For instance, symbols, units.

(5) In Table.3, the corrected deviation range and mean, should correct the results to the same decimal places.

Author Response

Dear reviewer:

Thank you for your comments concerning our manuscript entitled “Research on Elevation Survey Method of Sea-crossing Bridge under Adverse Conditions” (ID: sustainability-1863664). Those comments are all valuable and very helpful for revising and improving our paper, as well as the important guiding significance to our researches. We have studied comments carefully and have made correction which we hope meet with approval. Revised portion are marked in red in the paper. The main correction in the paper and the responds to the comments are as following:

Point 1: Line 116, the English expression needs to be improved.

Response 1: Thank you so much for your careful check. The English expression of Line 116 has been improved in the revised manuscript. Thanks again for your valuable comment.

Point 2: The detailed description of the specific time frame, completed of each group of tests, should be added in the manuscript of section 2.4.2.

Response 2: We gratefully appreciate for your valuable suggestion. We have been added the detailed description of the specific time frame in the revised manuscript of section 2.4.2.

Point 3: In Fig.6 (a); (b), (c), (d), (e), (f), (g), "good survey conditions" denoted by the blue texts, should give the detailed description.

Response 3: We gratefully appreciate for your valuable comment. The detailed description of "good survey conditions" has been added in the line 322 and line 323 of the revised manuscript.

Point 4: The notes of relevant formulas should be perfected in section 3.2.1. For instance, symbols, units.

Response 4: Thank you so much for your careful check. The notes of relevant formulas have been perfected in the revised manuscript of section3.2.1.

Point 5: In Table.3, the corrected deviation range and mean, should correct the results to the same decimal places.

Response 5: Thank you so much for your careful check. We have corrected the results of the corrected deviation range and mean to the same decimal places in Table.3.

We tried our best to improved the manuscript and made some changes in the manuscript. These changes will not influence the content and framework of the paper. And here we did not list the changes but marked in red in the revised manuscript.

We appreciate for Reviewer warm work earnestly, and hope that the correction will meet with approval.

Once again, thank you very much for your comments and suggestions.

Reviewer 2 Report

The authors proposed a method of elevation survey for long distances using dynamic compensation in order to enhance surveying accuracy. The authors then investigated the feasibility and practicability of this method using an outdoor experiment. The results of their work indicated that the proposed method is effective and can be especially useful where harsh environments reduce the accuracy of surveying. In the reviewer's opinion, the paper is interesting; however, there are a couple of comments that need to be addressed before it can be considered for publication. Please see my comments below:

1-       The importance and novelty of this study need to be emphasized more. Please provide more discussion on the necessity and novelty of this paper.

2-       There are some paragraphs that are confusing or hard to understand for readers. Please revise these paragraphs. Some include:

·         Page 2, paragraph 2.

·         Page 8, lines 272-281.

·         Etc.

A grammar and editorial review are recommended to polish the manuscript.

3-       Please provide a flow chart for section 2.4.2 (lines 221-238) as well. This could help readers to understand the steps better,

4-       It is suggested the introduction of the paper be strengthened by considering the following new article:

·         "Non-Destructive Testing Applications for Steel Bridges." Applied Sciences 11, no. 20 (2021): 9757.

·         "Non-destructive testing applications for in-service FRP reinforced/strengthened concrete bridge elements." In Nondestructive Characterization and Monitoring of Advanced Materials, Aerospace, Civil Infrastructure, and Transportation XVI, vol. 12047, pp. 59-74. SPIE, 2022.

5-       Please provide higher quality for figures 1 and 2.

Author Response

Dear reviewer:

Thank you for your comments concerning our manuscript entitled “Research on Elevation Survey Method of Sea-crossing Bridge under Adverse Conditions” (ID: sustainability-1863664). Those comments are all valuable and very helpful for revising and improving our paper, as well as the important guiding significance to our researches. We have studied comments carefully and have made correction which we hope meet with approval. Revised portion are marked in red in the paper. The main correction in the paper and the responds to the comments are as following:

Point 1: The importance and novelty of this study need to be emphasized more. Please provide more discussion on the necessity and novelty of this paper.

Response 1: Thank you for your comment. In the revised manuscript, the importance and novelty of this study have been emphasized and more discussion on the necessity and novelty of this paper which has been provided. We hope that our work can provide a clearer explanation of the principle and function of the elevation survey method proposed in this paper.

Point 2: There are some paragraphs that are confusing or hard to understand for readers. Please revise these paragraphs. Some include:  Page 2, paragraph 2. Page 8, lines 272-281. Etc. A grammar and editorial review are recommended to polish the manuscript.

Response 2: We gratefully appreciate for your valuable comment. We have revised the contents of “Page 2, paragraph 2. Page 8, lines 272-281. Etc.” and polished the manuscript by grammars and editorial reviews. We feel sorry for our carelessness.

Point 3:  Please provide a flow chart for section 2.4.2 (lines 221-238) as well. This could help readers to understand the steps better.

Response 3: We gratefully appreciate for your valuable comment. We have provided the flow chart about the content of section 2.4.2. Thanks again for your valuable comment.

Point 4:  It is suggested the introduction of the paper be strengthened by considering the following new article:

 "Non-Destructive Testing Applications for Steel Bridges." Applied Sciences 11, no. 20 (2021): 9757.

 "Non-destructive testing applications for in-service FRP reinforced/strengthened concrete bridge elements." In Nondestructive Characterization and Monitoring of Advanced Materials, Aerospace, Civil Infrastructure, and Transportation XVI, vol. 12047, pp. 59-74. SPIE, 2022.

Response 4: Thank you for your valuable suggestion. We have cited related articles in paragraph 1 on page 2 of the revised manuscript.

Point 5:  Please provide higher quality for figures 1 and 2.

Response 5: Thank you so much for your careful check. We have provided higher quality for related figures.

We tried our best to improve the manuscript and made some changes in the manuscript. These changes will not influence the content and framework of the paper. And here we did not list the changes but marked in red in the revised manuscript.

We appreciate for Reviewer warm work earnestly, and hope that the correction will meet with approval.

Once again, thank you very much for your comments and suggestions.

Round 2

Reviewer 2 Report

The authors addressed the comments adequately. Accordingly, the paper can be accepted.